# Analysing the Influence of WHO Initiatives on the Scientific Discourse of Noncommunicable Diseases through a Bibliometric Approach

**DOI:** 10.3390/ijerph20186714

**Published:** 2023-09-06

**Authors:** Ana Teresa Santos, Cátia Miriam Costa, Luisa Delgado-Márquez, Raquel Maria Banheiro

**Affiliations:** 1Egas Moniz Center for Interdisciplinary Research, Egas Moniz School of Health & Science, 2800 Almada, Portugal; 2Centro de Estudos Internacionais, Instituto Universitário de Lisboa (ISCTE-IUL), 1649-026 Lisbon, Portugal; catia.miriam.costa@iscte-iul.pt; 3Department of Applied Economics, University of Granada, 18071 Granada, Spain; luisadm@go.ugr.es; 4Business Research Unit (BRU-IUL), Instituto Universitário de Lisboa (ISCTE-IUL), 1649-026 Lisbon, Portugal; 5Nova Medical School, Universidade Nova de Lisboa, 1099-085 Lisbon, Portugal; raquel.m.banheiro@edu.nms.unl.pt

**Keywords:** non-communicable diseases, bibliometrics, discourse analysis, United Nations, World Health Organization, health diplomacy

## Abstract

Noncommunicable diseases (NCDs) present a major public health challenge, prompting their inclusion in the United Nations’ 2030 Agenda for Sustainable Development. In response, the World Health Organization (WHO) has implemented various initiatives, including a comprehensive monitoring framework with global targets and indicators. However, the extent to which these initiatives have shaped the scientific discourse remains unclear. This article addresses this knowledge gap through a two-fold approach. Firstly, a bibliometric analysis of 14,187 studies spanning over 60 years is conducted, identifying key contributors and trends. Secondly, the content analysis compares these trends to the goals established by the WHO. The findings indicate that the WHO initiatives have accelerated scientific research, and elevated global targets and indicators as central themes in scholarly discussions, since 2011. This study takes an innovative approach that contributes to the advancement of knowledge in this field, by providing valuable insights into the impact of WHO initiatives on the scientific debate surrounding NCDs, and offering guidance for policymakers, researchers, and stakeholders engaged in combating these diseases.

## 1. Introduction

Noncommunicable diseases (NCDs) represent a significant public health challenge in the twenty-first century [1]. The combination of genetic, physiological, environmental, and behavioural factors influence these long-lasting medical conditions [1]. According to the World Health Organization (WHO), NCDs are the leading cause of global mortality, accounting for 74% of all annual deaths (World Health Organization, 2023). Among the most prevalent NCDs are cardiovascular diseases (17.9 million deaths annually), cancers (9.3 million), respiratory diseases (4.1 million), and diabetes (2.0 million) [1].

The recognition of NCDs as a significant challenge for sustainable development is evident in the 2030 Agenda for Sustainable Development. Heads of state and government have committed to ambitious national responses to reduce premature mortality from NCDs by one-third through prevention and treatment (SDG target 3.4). As a multilateral United Nations body, the WHO is essential in the development of health diplomacy targeting intergovernmental solutions for global health challenges. Therefore, the WHO was crucial in coordinating and promoting global efforts to combat NCDs, and achieve SDG target 3.4 through international collaboration [1].

In 2011, the WHO adopted the UN General Assembly’s Political Declaration on NCDs, followed by the adoption of a mortality target in May 2012 to reduce premature mortality from NCDs by 25% by 2025. Concurrently, the WHO developed a comprehensive global monitoring framework, consisting of nine global targets and twenty-five indicators endorsed by Member States during the World Health Assembly in May 2013 [2]. This framework aimed to assess global progress in preventing and controlling the four major NCDs and their primary risk factors.

Member States were encouraged to develop national NCD targets and indicators based on the global framework. The nine voluntary global targets focused on reducing the global mortality from the four main NCDs, addressing leading risk factors, and strengthening national health system responses. Additionally, the framework aimed to drive progress in NCD prevention and control, raise awareness, reinforce political commitment, and promote global action to combat these deadly diseases, while advancing the three dimensions of sustainable development: economic development, environmental sustainability, and social inclusion. On their part, the WHO has conducted a series of global country capacity surveys, to assess how each member state has implemented these policies [3]. The findings have been reported in the 2015, 2017, 2020, and 2022 NCD progress monitor reports, each pertaining to data collected in the preceding year.

More recently, in 2017, the WHO advocated for a range of “best buy” measures to mitigate the impact of NCDs, including taxation, marketing limitations, and product labelling. Implementing these measures, especially in low- and middle-income countries, could yield substantial healthcare savings and productivity improvements, of up to USD 230 billion, by 2030 [4].

Considering the momentum generated by these WHO efforts, assessing their impact on the scientific literature and the trajectory followed by publications over the years is crucial to gaining insights into the evolving discussions on NCDs. This paper applies a comprehensive bibliometric and content analysis to quantitatively analyse scientific studies published on NCDs since 1960. By examining the document content and author information, we aim to identify the journals and authors contributing the most to NCD research, and explore the trending topics within the field. To the best of our knowledge, this is the first study utilising a bibliometric approach to analysing the production of the scientific research on NCDs over time.

## 2. Materials and Methods

### 2.1. Methodology

Bibliometrics is a quantitative method for analysing and measuring the scientific literature, such as books, articles, and publications, offering a structured approach for evaluating research outcomes and their effects, identifying developing trends, and charting the intellectual landscape of a particular field [5]. The bibliometric method helps provide objective, reliable, and structured point-in-time analysis of trends, shifts in disciplines, research themes, and most productive authors and author institutions, presenting a broader picture of the field research through time. Researchers believe that the employment of bibliometric methodology is a convenient means of exploring patterns within research fields, including themes, conceptual frameworks, and intellectual organization [6]. Research studies in various fields have used bibliometric analysis to evaluate developments in specific specialities, such as radical wireless technologies [7], innovation studies [8], and sustainability-focused service innovation [9], to identify emerging trends.

### 2.2. Data Collection

The academic literature published in series plays a crucial role in understanding the development of intellectual pursuits. It offers an essential perspective, by providing comprehensive and quality-controlled information that sheds light on various facets of knowledge production. The database constructed for this study encompasses several deliberate choices, consolidation procedures, and validation efforts, which are essential to report. The schematic representation of our expansive data retrieval approach, intricately illustrated in Figure 1, encapsulates the essence of this bibliometric study.

Commencing our methodology, we initiated the gathering of scientific discourse from the comprehensive Scopus database, guided by references such as [10,11]. The selection of Scopus stems from its stature as the largest citation and abstract repository for peer-reviewed literature, equipped with tools tailored for executing scientometric analyses, as acknowledged by [12,13,14]. This expansive database encompasses a multitude of journals, and affords the extraction of diverse attributes, including authorship and content. Subsequently, our search strategy was crafted to encompass articles incorporating the terms “non-communicable” and “diseases” in their title, keywords, or abstract, and restricted to publications between 1960 and 2022. The decision to conclude our search in 2022 was driven by its provision of the most comprehensive records. Thorough data extraction followed, encapsulating article titles, author identities, journal titles, publication years, and affiliation particulars. In a judicious discernment, content forms such as books, book chapters, editorials, errata, letters, retracted articles, and succinct surveys were deliberately excluded. Notably, proceedings from single conferences were also omitted, due to their absence of longitudinal insight. The final step of our meticulous curation was the deliberate focus on a specific subset of documents, specifically articles and reviews, while categorically excluding alternative content genres, such as editorials or book reviews. This systematic pursuit culminated in the unearthing of 14,187 distinctive documents ensconced within 3201 distinct source titles, comprising a compilation of 10,993 papers and 3194 reviews.

### 2.3. Text Manipulation

The selected abstracts were converted electronically, and saved in rich text format (RTF). To prepare the set texts for analysis, we carried out a proofreading process that encompassed spell-checking and comprehensive searching, to ensure the inclusion of all content. Subsequently, we proceeded with the pre-processing stage, which involved four key steps: text segmentation, the elimination of numbers and punctuation, conversion to lowercase, and the removal of stop words. Text segmentation, also referred to as tokenisation, involves the division of the original text into individual words based on word boundaries, such as white spaces [15]. We then converted all text to lowercase, and eliminated numbers, punctuation, and running heads. Finally, we discarded insignificant or auxiliary words from the end of word pieces, to focus on words that carried meaningful content. These words, commonly known as “stop words”, were removed, as they did not contribute to the analysis. Examples of stop words that were eliminated from documents include “for”, “to”, “was”, “and”, “the”, “of”, and “by”. The removal of stop words enhanced the efficiency of the software, without impacting the results of the text analysis [16]. All the analyses were performed using Rstudio software, version 4.2.3. Additional packages were also used, such as “tm”, “dplyr”, “ggplot2”, “igraph”, “RTextTools”, and “Snowballc”.

## 3. Results

### 3.1. The Generation of Scientific Discussions

#### 3.1.1. The Speed of Production

The interest in NCDs is expanding, and can be seen from various viewpoints. Multiple literature revisions showed that several studies have been published on this topic over the years, not only on NCDs themselves [17,18], but also on the effects of lifestyle changes on NCDs [19], socioeconomic inequalities [20], and the development of new indicators [21]. Aside from looking at longer and shorter periods, the published evaluations looked at various databases primarily covering health and social science, while sectors such as engineering were left out, and could potentially be a source of additional data [22]. In this study endeavour, we did not limit the scientific topic, period, or publication, so that we could evaluate as many articles as feasible. Thus, Figure 2 shows the number of articles published about “non-communicable diseases” since 1960.

These findings demonstrate a significant increase in NCD-related papers after 2011, following decades of stable scholarly productivity. This pattern shift could have been aided by the UN General Assembly’s adoption of the Political Declaration on Noncommunicable Diseases in 2011. This year, the WHO devised a worldwide monitoring framework to enable the global tracking of progress in preventing and controlling major noncommunicable illnesses and their essential risk factors, including cardiovascular disease, cancer, chronic lung disease, and diabetes.

#### 3.1.2. Channels of Publication

From 1960 onwards, 3201 source titles have published articles concerning “non-communicable diseases”. However, a significant portion of these sources, precisely 1670, have yet to publish one article each over 60 years. Additionally, 526 sources have published merely two manuscripts. To visually depict the primary publication venues for these topics throughout the years, Figure 3 was created. We only included in this analysis journals that published 62 articles or more between 2000 and 2022 (equivalent to an average of at least one paper per year). As a result, we considered 17 journals for this purpose.

PLoS ONE was the journal publishing the most papers, with 518 registered, although these articles were focused between 2008 and 2022. The BMC Public Health had the second highest number, with 494 references dating back to 2004. Over 11 years, the International Journal of Environmental Research and Public Health received 338 papers. The journals BMJ Open and Nutrients published 294 and 280 publications, respectively. All of the others had fewer than 150 articles.

#### 3.1.3. Geographies

Regarding the number of nations represented in the authors list, we reviewed all affiliations provided, and discovered writers in 183 countries/territories. We show the distribution of authorships by continent and nation in Figure 4. The darker the colour, the larger the number of authors linked with national institutions (we did not find the grey territories in the author list). Table 1 shows the proportions in detail.

Even though the United States has the most significant number of authors writing about NCDs, the top 10 authors who publish the most are not exclusively from the US. In fact, only one author, Christopher J. L. Murray, is affiliated with a US institution: the Institute for Health Metrics and Evaluation at the University of Washington in Seattle.

Among the top ten authors, six are affiliated with Iranian institutions: Farshad Farzadfar from the Non-Communicable Diseases Research Center and Endocrinology and Metabolism Population Sciences Institute at Tehran University of Medical Sciences; Roya Kelishadi from the Pediatrics Department and Child Growth and Development Research Center at the Research Institute for Primordial Prevention of Non-communicable Disease, Isfahan University of Medical Sciences; Farid Najafi from the Research Center for Environmental Determinants of Health at the School of Public Health, Kermanshah University of Medical Sciences, Mostafa Qorbani from the Non-Communicable Diseases Research Center, Alborz University of Medical Sciences; Reza Malekzadeh from the Liver, Pancreatic, and Biliary Diseases Research Center at the Digestive Diseases Research Institute, Tehran University of Medical Sciences, and Bagher Larijani from the Endocrinology and Metabolism Research Center and Endocrinology and Metabolism Clinical Sciences Institute at Tehran University of Medical Sciences.

Among the top 10 authors, Alan D. Lopez represents Australia, and is affiliated with the School of Population and Global Health at The University of Melbourne. Deborah Carvalho Malta, on the other hand, hails from Brazil, and is associated with the Federal University of Minas Gerais, School of Nursing, Department of Maternal and Child Nursing and Public Health in Belo Horizonte. Lastly, Martin Mckee, from the United Kingdom, is affiliated with the Faculty of Public Health and Policy at the London School of Hygiene and Tropical Medicine—specifically, the Centre for Global Chronic Conditions in London.

### 3.2. The Contents Published

#### 3.2.1. Thematic Focus of the Field of NCDs

We investigated research publications on NCDs to identify themes that dominate the field. Words were segmented to their roots and stems, to ensure that the same concepts were considered together, regardless of being female/male or singular/plural. The frequency of the words was analysed through a word cloud technique. The visuals are set up to facilitate understanding regarding the trending topics: the larger the font size and the darker the colour, the higher the frequency of occurrence. Figure 5 suggests that the most frequently used words include “cancer”, “diabetes”, “obesity”, and “hypertension”.

This representation highlights stems that follow the targets and indicators outlined in the framework accepted by Member States at the World Health Assembly (WHA) in May 2013. For example, in terms of mortality and morbidity, the goal was to reduce the overall mortality from cardiovascular diseases, cancer, diabetes, or chronic respiratory diseases by 25%. We discovered related stems in the word cloud, derived from words such as “mortal”, “death”, “cancer”, and “diabetes” themselves, as well as “cvd” (from cardiovascular disease), related to cardiovascular diseases, or asthma as a chronic respiratory disease.

Such findings may imply that the WHO framework motivates research efforts, which are then translated into scientific discourse. When the framework was created, the purpose was to drive progress in NCD prevention and control, and offer a foundation for campaigning, raising awareness, reinforcing political commitment, and advocating global action to combat these terrible diseases. This framework was also intended to aid in formulating a new development agenda that promotes the three pillars of sustainable development: economic development, environmental sustainability, and social inclusion. The scientific discourse indicates that researchers accepted the challenge, and are working towards that aim.

#### 3.2.2. The Most Frequent Diseases/Conditions

Since 1960, the 14,187 articles that we found addressed multiple diseases with different frequencies. The ones occurring more than 1000 times in the abstracts were selected, and are shown in Figure 6. Although these diseases are labelled with a single word, for some of them, the figures illustrated resulted from the collection of terms considered clinically similar. For example, “cardiovascular” includes the frequencies of “CVD” (cardiovascular disease), “infarction”, and “cardiovascular” itself. For both “diabetes” and “hypertension”, preconditions are also considered, such as “prediabetes” and “prehypertension”, respectively. Regarding “Tobacco”, multiple expressions are also considered: “nicotine”, “cigarette”, “smoke”, and “tobacco” itself.

These words and expressions are in line with the framework defined by the WHO, encompassing both targets and indicators. Several articles examine the links between problematic alcohol use and the risk of developing hypertension [23], diabetes, asthma, and cardiovascular disease [24]. This discourse is consistent with the WHO’s aim of reducing the harmful use of alcohol by at least 20%. The rising usage of “Mortality” also reflects the target of reducing 25% overall mortality from the top NCDs, as multiple articles discuss it in relation to cardiovascular diseases, cancer, diabetes, or chronic respiratory diseases [25,26,27]. Among them, the first three (cardiovascular diseases, diabetes, and obesity) are the ones being addressed the most. Socioeconomic inequalities are being discussed as essential factors in preventing diabetes [28], with ultra-processed food being identified as increasing the risk of obesity and cardiovascular disease [29], and physical inactivity being associated with a substantial global burden [25]. This rising interest could be associated with the established targets of (1) halting the rise in diabetes and obesity, and (2) reducing by 15% the prevalence of insufficient physical activity. Hypertension was identified as a leading risk factor for strokes [30], and advances in the management of hypertensive patients were also discussed, such as the development of non-intrusive and clinically validated devices for ambulatory blood pressure (BP) measurement, and the consideration of drugs initially developed for conditions different from hypertension, including heart failure and diabetes, have been demonstrated to lower BP significantly [31]. This concern is being addressed by the WHO targets of reducing 30% of the mean population intake of salt/sodium, and achieving a 25% reduction in the prevalence of raised blood pressure. Finally, multiple articles discuss the WHO’s recommendation of policies to reduce by 30% the consumption of tobacco products, although the regulation has been strongly opposed by the affected industries, which requires cooperation between disciplines.

### 3.3. The Networks in NCDs

This particular subsection delves further into the more complex scenario of the corpus, comprising the articles related to NCDs. We accomplished this analysis by examining association maps created via unigram analysis, based on the words extracted from the abstracts. In semantic network analysis, the relationships between various terms are empirically defined, allowing us to disentangle the context from the word structure [32]. The lines depicted on the following graphs in this section symbolise the co-occurrence of nodes, thereby establishing connections between them. The greater the thickness, the greater the number of times the word is repeated.

Figure 7 shows that specific cancer types are discussed (such as liver, colon, breast, skin, and lung), while other directions are also followed in the scientific literature. For instance, “screening”, “diagnosis”, “treatment”, and “prevention” strategies are discussed, as well as the “mortality” and “incidence”. Special consideration is also given to “registries” as a strategy to map the disease, and follow the development and “research” to find new disease management and treatment strategies. Only words repeated 16 times or more were retrieved.

The main focus of discussion concerning the topic of “cardiovascular” is revealed in Figure 8. The primary emphasis is placed on identifying “risk” factors that contribute to the development of cardiovascular disease. Furthermore, certain factors are examined in greater depth, including “diabetes”, “hypertension”, “obesity”, and “atherosclerotic” conditions. Notably, significant attention is given to the consequences of cardiovascular disease, specifically “mortality” and “deaths”. The scientific discourse also strongly emphasises looking towards the “future” and exploring strategies for “prevention”. Only words repeated 12 times or more are displayed.

The primary subjects related to “diabetes” are depicted in Figure 9. The most frequently mentioned words in this context are “mellitus,” which refers to the disease itself, and “gestational”, which pertains to a condition occurring during pregnancy. In the gestational stage, a hormone called human placental lactogen hinders the effective use of insulin in the body. Additionally, the literature discusses various “complications” associated with diabetes, including “stroke,” as well as “risk” factors, such as “hypertension” and “obesity”. Moreover, the scientific literature raises concerns regarding “undiagnosed” diabetes. Only words repeated 21 times or more are displayed.

The discourse surrounding “hypertension” is depicted in Figure 10. The word “diabetes” is the most frequently used as a risk factor for the illness, followed by “obesity”, “dislipidemia”, and “smoking”. Furthermore, special emphasis is placed on investigating the disease’s “prevalence,” “management,” “control”, and “treatment.” Notably, there is a strong emphasis on “undiagnosed” and “uncontrolled” hypertension. The debate also focuses on “patients” and “self-reported” results. Only words repeated 19 times or more are displayed.

Regarding “overweight”, the scholarly literature, illustrated in Figure 11, focuses on examining “individuals”, particularly “children” and those in their “childhood”. It also delves into discussions about “weight” and “BMI” (body mass index), as well as individuals categorised as “obese”, and the concept of “obesity” itself. Notably, the term most frequently associated with “overweight” is “obesity”, which is a long-term consequence. Conversely, the topic of “underweight” is also addressed. The primary motivation for articles on this subject is often the “prevalence” of the issue. Only words repeated 9 times or more are displayed.

The scientific literature explores the topic of “smoking” from various angles, as shown in Figure 12. It not only covers the act of “consumption” itself, and the use of “tobacco” or “cigarettes”, but also focuses on the “prevalence” of smoking, and the associated “habit”. Additionally, the literature addresses certain coexisting conditions, such as “obesity” and “hypertension”, and recent shifts in national policy regarding commercial interests [33]. Furthermore, the literature discusses strategies aimed at “reducing”, “quitting”, or achieving the “cessation” of smoking. Notably, a significant association is observed between “smoking” and the habit of consuming “alcohol”. Only words repeated 13 times or more are displayed.

## 4. Conclusions

In the landscape of twenty-first century public health, NCDs stand out as significant challenges, prompting the WHO to spearhead numerous initiatives against these ailments. Nonetheless, the task of presenting nonbiased and potentially critical analysis within the ongoing healthcare discourse concerning NCDs is riddled with challenges. These include: (1) the presence of data gaps that impede comprehensive assessments and meaningful conclusions; (2) the presence of financial interests linked to specific treatments, which could impact funding, partnerships, and the analysis’s credibility; (3) the intricate nature of NCDs, encompassing medical, social, economic, and cultural aspects, necessitating a holistic grasp of diverse disciplines; and (4) the sway of political considerations that can affect agendas, posing difficulties in garnering acceptance for impartial insights.

In this work, we undertook a comprehensive bibliometric and content analysis of the scientific research on NCDs, to understand how WHO initiatives impacted multiple decades of publications, and whether they drove the scientific discourse. The timing of this study is crucial, because it has been ten years since the WHO recommended a set of population-level strategies to combat NCDs in 2013, which was endorsed by all 194 member states.

This study looked at 14,187 studies from the Scopus database that covered over 60 years, and gained valuable insights into the evolving NCD debate. Firstly, the scientific literature on NCDs has experienced a substantial growth and accelerated pace since adopting the WHO’s Political Declaration on NCDs in 2011. This growth indicates the increasing attention and recognition given to NCDs as a significant public health concern. Secondly, our analysis identified the top journals (PLoS ONE, BMC Public Health, International Journal of Environmental Research and Public Health, BMJ Open, and Nutrients), which published at least 280 articles during the 60 years considered, and the top authors, who proved to be affiliated not only with the US and UK, but mainly from Iranian institutions. These key contributors play a crucial role in shaping the scientific discourse, and advancing knowledge in the field. By identifying these influential sources, researchers, policymakers, and stakeholders can prioritise collaborations and dissemination efforts, to maximise the impact of their work. Thirdly, we observed that the trending topics in the NCD research align closely with the goals and targets established by the WHO. These findings indicate that the WHO’s efforts in promoting NCD prevention and control have resonated within the scientific community. The research literature consistently focused on reducing mortality and addressing risk factors, suggesting a solid alignment between scientific priorities and global health objectives.

Overall, our findings highlight the impact of the WHO’s initiatives in shaping the scientific literature on NCDs. The WHO’s comprehensive global monitoring framework, comprising targets and indicators, has influenced the research direction, and fostered a collaborative and coordinated approach to addressing NCDs. By bridging the gap between research and policy, our study provides valuable insights for future research and policy formulation in NCD prevention and control. The findings can inform evidence-based decision-making, resource allocation, and the development of targeted interventions to combat NCDs effectively.

## 5. Limitations

While this study has made significant contributions, it is crucial to recognise its inherent limitations, which, in turn, create promising avenues for potential future research. Our analysis was primarily centred on English-language publications, potentially confining our perspective to a subset of the comprehensive global research landscape concerning NCDs. It is widely acknowledged that research findings from non-English-speaking researchers could yield divergent statistical outcomes (as evidenced by [34]). This stems from the tendency for positive or statistically significant results to be favoured in English publications, leading to what is termed “English-language bias” [35] or the “Tower of Babel” bias [36].

In our study, we unearthed 1341 manuscripts in languages other than English, offering a potential wealth of additional information regarding the ways in which NCDs are approached from the standpoint of non-English-speaking authors. Incorporating such a substantial body of content into the existing portrait of the scientific discourse would be advantageous.

Importantly, we refrained from examining or comparing the research output of highly productive institutions with that of the WHO. Furthermore, we did not juxtapose the scientific dialogues surrounding the WHO with its officially issued statements. Looking ahead, continuous vigilance towards, and assessment of, research outputs are indispensable in ensuring that scientific endeavours remain attuned to the evolving complexities posed by NCDs. This sustained evaluation will ultimately contribute to enhancing the prevention, control, and management of these diseases on a global scale.

## Figures and Tables

**Figure 1 ijerph-20-06714-f001:**
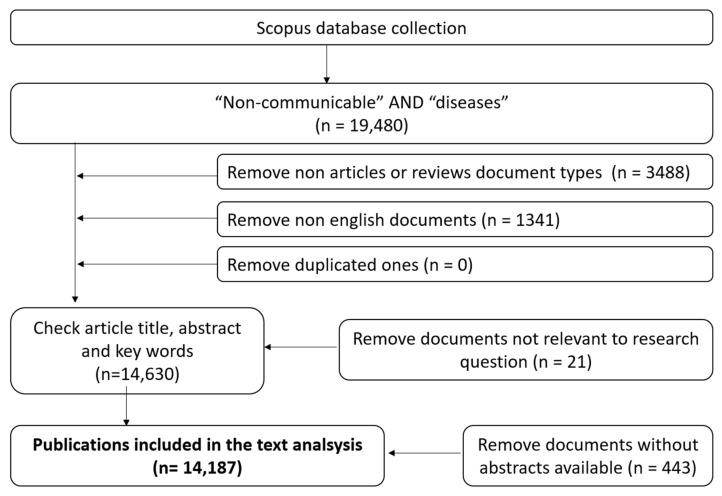
Flowchart of the literature identification and selection for this study.

**Figure 2 ijerph-20-06714-f002:**
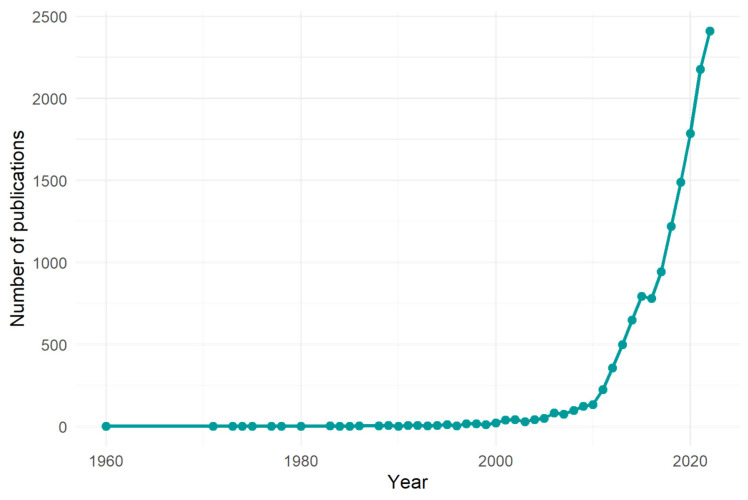
Number of papers published per year with the expression “non-communicable diseases”.

**Figure 3 ijerph-20-06714-f003:**
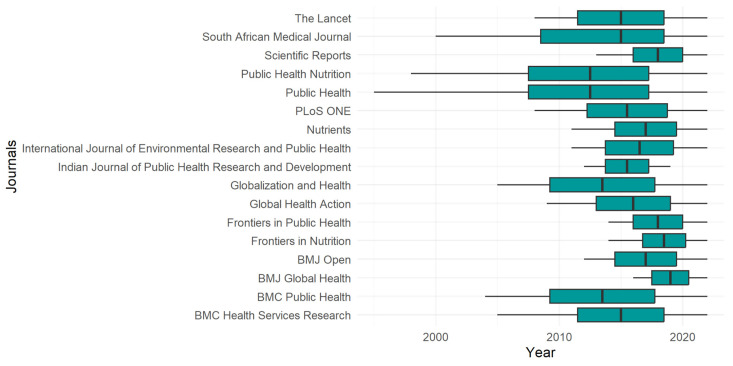
The distribution of the articles published by source title through the years.

**Figure 4 ijerph-20-06714-f004:**
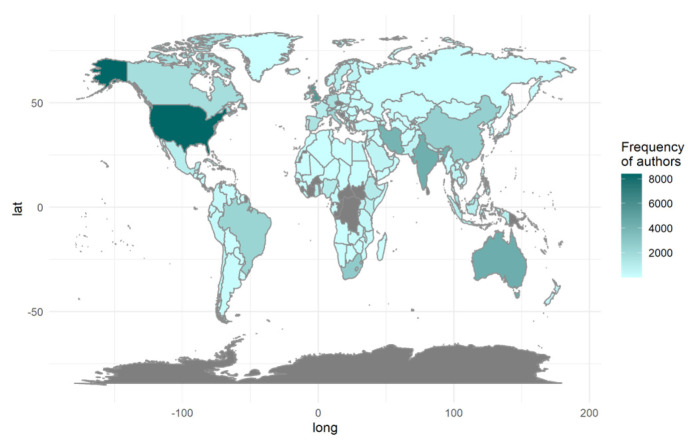
The locations of the authors (the darker the colour, the higher the number of authors).

**Figure 5 ijerph-20-06714-f005:**
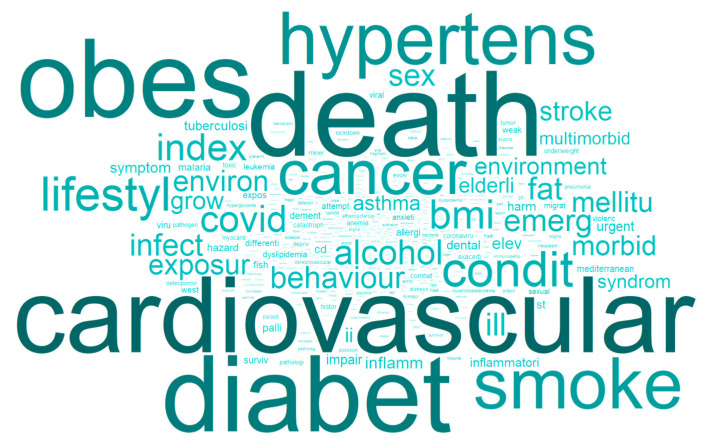
Word cloud visualisation of frequently and repeatedly used words.

**Figure 6 ijerph-20-06714-f006:**
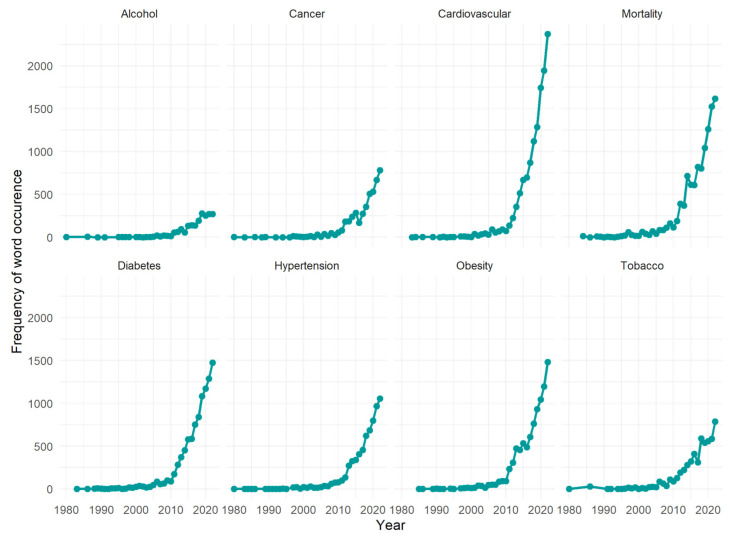
The occurrence of the most frequent diseases/conditions.

**Figure 7 ijerph-20-06714-f007:**
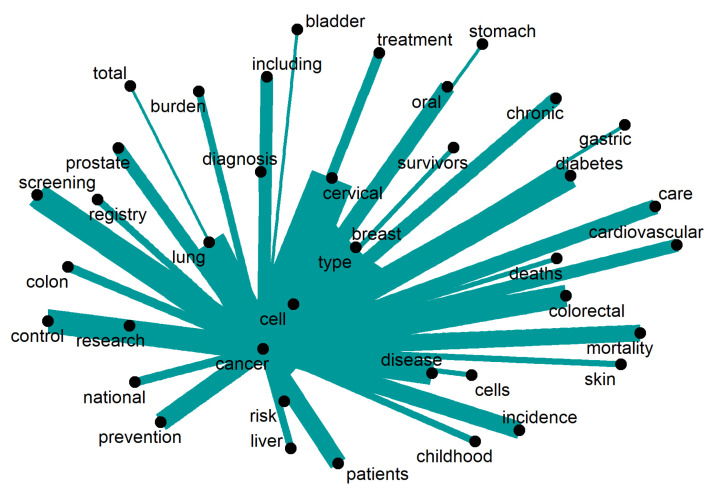
Words around “cancer”.

**Figure 8 ijerph-20-06714-f008:**
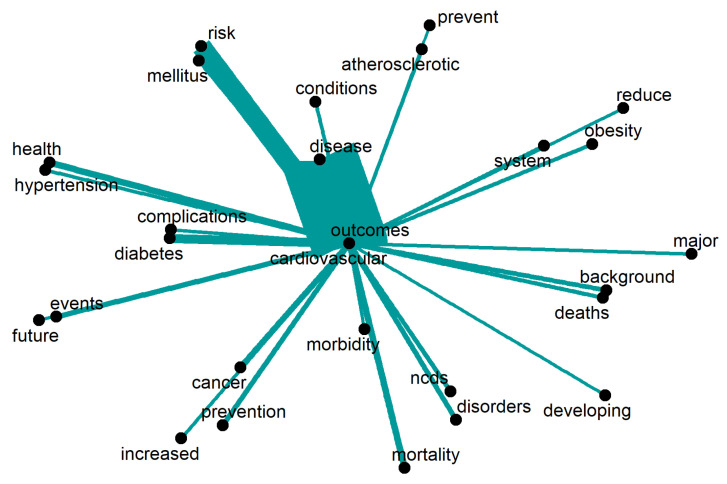
Words around “cardiovascular”.

**Figure 9 ijerph-20-06714-f009:**
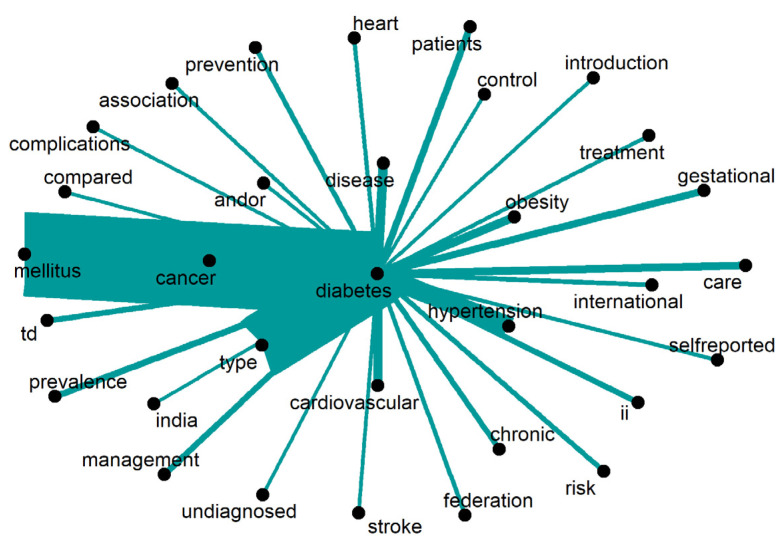
Words around “diabetes”.

**Figure 10 ijerph-20-06714-f010:**
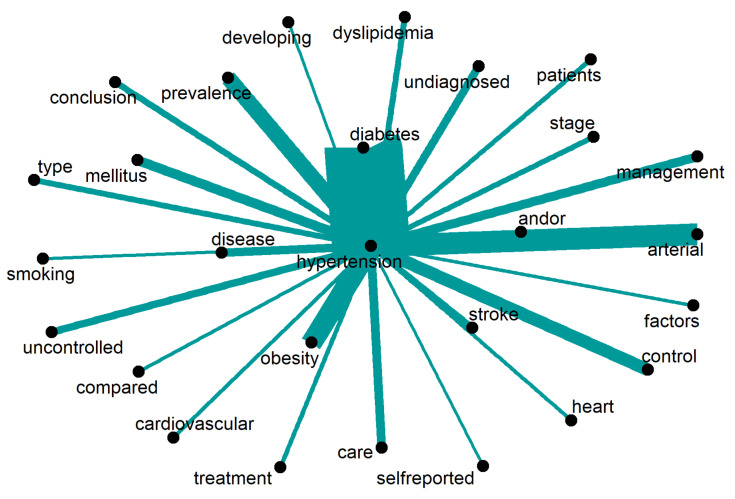
Words around “hypertension”.

**Figure 11 ijerph-20-06714-f011:**
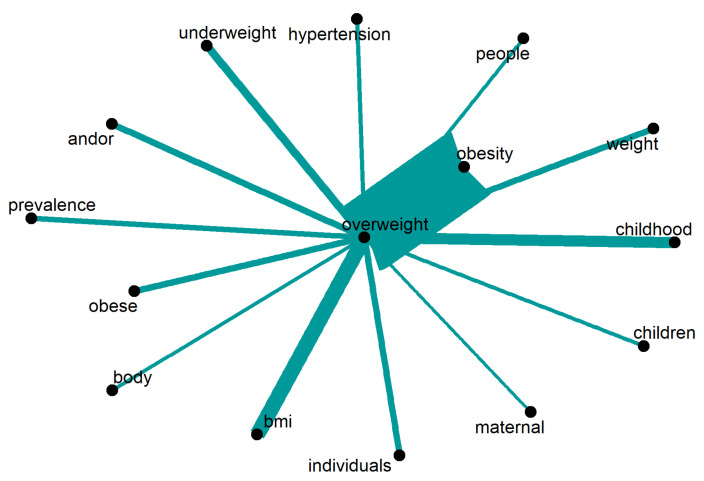
Words around “overweight”.

**Figure 12 ijerph-20-06714-f012:**
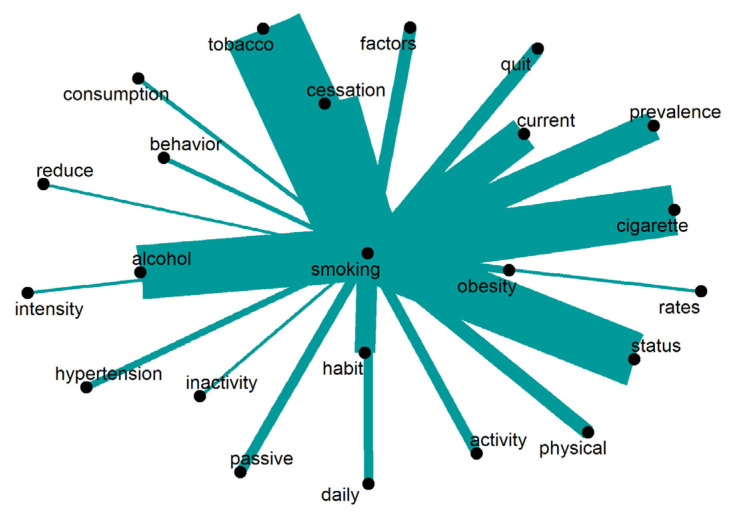
Words around “smoking”.

**Table 1 ijerph-20-06714-t001:** Distribution of authors by continent.

Continent	Proportion
Africa	14%
Asia	29%
Europe	28%
North America	16%
Oceania	7%
South America	5%

## Data Availability

The datasets generated and analyzed during the current study are available from the corresponding authors on reasonable request.

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
