# Peer review of "Analysing the Influence of WHO Initiatives on the Scientific Discourse of Noncommunicable Diseases through a Bibliometric Approach"

_ijerph, 2023, doi:10.3390/ijerph20186714_

Round 1
Reviewer 1 Report
There is no Discussion section between Results and Conclusion. Please add it back.
Please report which software and the methodological details on how to use the software (e.g. with which settings and parameters) to produce the figures.
Figure 5: The word cloud contains many truncated words, e.g. obes, hypertens, diabet... these are not words.
The title says "the influence of WHO..." That's very good and novel. However, nothing in the results was answering that research question. WHO was not among the most productive institutions. WHO was not among the most frequently appearing words from the analyzed papers. Basically, the reviewer cannot see the influence of WHO based on the results presented. Either present more convincing and results directly related to WHO, or change the title to remove the mentioning of WHO.
Limitations of the study should be reported.
Author Response
There is no Discussion section between Results and Conclusion. Please add it back.
We express our gratitude to the reviewer for their suggestion. In response, we have taken heed of Reviewer 2's advice to underscore the intricacies involved in presenting an unbiased and potentially critical analysis within the context of contemporary healthcare discussions.
Please report which software and the methodological details on how to use the software (e.g. with which settings and parameters) to produce the figures.
We appreciate the raised point. In fact, all the analyses were conducted using R software, specifically version 4.2.3. We also employed supplementary packages including "tm", "dplyr", "ggplot2", "igraph", "RTextTools" and "Snowballc." (see section 2.3)
Figure 5: The word cloud contains many truncated words, e.g. obes, hypertens, diabet... these are not words.
We concur with and are adhering to the reviewer's feedback. Following the stemming process, we are now dealing with "stems" instead of "words," as the expressions have been shortened to exclude gender or number distinctions like female/male or singular/plural forms. We updated the section 3.2.1 accordingly.
The title says "the influence of WHO..." That's very good and novel. However, nothing in the results was answering that research question. WHO was not among the most productive institutions. WHO was not among the most frequently appearing words from the analyzed papers. Basically, the reviewer cannot see the influence of WHO based on the results presented. Either present more convincing and results directly related to WHO, or change the title to remove the mentioning of WHO.
We extend our gratitude to the reviewer for the insightful comment and the chance to provide further clarification. The global targets and indicators, established by the WHO, serve as the foundation for our assessment of their translation into subsequent scientific literature. However, it's important to note that this study does not delve into an examination of the most prolific institutions, nor does it explore the discourse surrounding WHO. We acknowledge this limitation in our conclusion and recommend addressing it in future research endeavors (see lines 417-423).
Limitations of the study should be reported.
We value the suggestion provided. Building upon the earlier comment, we have included a new section on the limitations inherent to this study. This includes our omission of a comparison involving the most prolific institutions, as well as the absence of an exploration into the scientific discussions surrounding the WHO within our analysis (see lines 400-423).
Reviewer 2 Report
Comments on “Analysing the Influence of WHO Initiatives on the Scientific Discourse of Noncommunicable Diseases through a Biblio-I metric Approach’
General Comments:
1. This is a well written and well-presented paper highlighting evidence on the topic of WHO’s initiatives on addressing communicable diseases. The methodology of the research is well explained. The outcomes and evidence are clearly provided.
2. I have one major concern. The authors (who come from Portuguese institutions, and I suspect are not native English speakers) choose to review evidence only in the English language journals. I recognize that these journals are some of the most respected in the field. However, there is a growing concern and evidence that important findings and contributions are not getting the recognition they need due to the fact their evidence is not in English.
I think this paper should not be published without a somewhat in-depth discussion of the challenges of publication for those who do not use English and/or have the ability to get the information and experience of presenting their findings.
3. This paper does not need to provide a discussion about how to address this issue. It does need to highlight it as a challenge to presenting nonbiased and quite possibly critical analysis to the present day health care discussion.
Author Response
This is a well written and well-presented paper highlighting evidence on the topic of WHO’s initiatives on addressing communicable diseases. The methodology of the research is well explained. The outcomes and evidence are clearly provided.
We express our gratitude to the reviewer for their comprehensive review and feedback.
I have one major concern. The authors (who come from Portuguese institutions, and I suspect are not native English speakers) choose to review evidence only in the English language journals. I recognize that these journals are some of the most respected in the field. However, there is a growing concern and evidence that important findings and contributions are not getting the recognition they need due to the fact their evidence is not in English. I think this paper should not be published without a somewhat in-depth discussion of the challenges of publication for those who do not use English and/or have the ability to get the information and experience of presenting their findings.
We are grateful to the reviewer for bringing up this important point. The issue of "English-language bias" is a longstanding concern that we have indeed acknowledged in our conclusion. This is evident in our exclusion of 1,341 articles due to their non-English language content. This particular limitation presents a valuable avenue for future research to incorporate and explore these overlooked findings as well (see lines 400-409).
This paper does not need to provide a discussion about how to address this issue. It does need to highlight it as a challenge to presenting nonbiased and quite possibly critical analysis to the present day health care discussion.
We extend our gratitude to the reviewer for their valuable input. In the concluding section (first paragraph), we outline four predominant challenges of distinct natures, as follows: financial interests, data insufficiency, the intricate nature of NCDs, and the impact of political considerations.
Reviewer 3 Report
1. Why did the author select Scopus as a database and not use other databases?
2. The Data collection and Data sorting are not clear.
3. Describe the limitation and straightness and suggestions for future study.
4. What are the home messages?
5. It could be better that lines 177-200 convert to the table.
6. The thickness lines legend of Figure 7-12 is unclear.
7. Does the author consider RTC study? Why did you not consider animal studies or others?
Author Response
Why did the author select Scopus as a database and not use other databases?
We appreciate the point raised. Scopus was chosen because it is the largest citation and abstracting database for peer-reviewed literature and has tools suitable for performing this type of scientometric analysis. We support this selection with three additional references in the text (see lines 105-108).
- Burnham, J.F. Scopus database: A review. Biomed. Digit. Libr. 2006, 3, 1–8.
- Schotten, M.; Meester, W.J.; Steiginga, S.; Ross, C.A. A brief history of Scopus: The world’s largest abstract and citation database of scientific literature. In Research Analytics; Auerbach Publications: Boca Raton, FL, USA, 2017; pp. 31–58.
- Borgohain, D.J. Research Output of Dibrugarh University: A Scientometric Analysis based on Scopus Database. Libr. Philos. Pract. E-J. 2020, 4827.
The Data collection and Data sorting are not clear.
We extend our gratitude to the reviewer for affording us the chance to enhance our work. We have taken the opportunity to revise the section, augmenting the description to offer greater clarity regarding the methodology employed for data collection and subsequent data organization (see section 2.2).
Describe the limitation and straightness and suggestions for future study.
We value the suggestion and have incorporated a dedicated limitations section at the conclusion of the manuscript. This section addresses the primary strengths and weaknesses of our study, notably acknowledging the constraints of focusing solely on English-language publications. This approach resulted in the omission of insights from over 1300 papers, indicating a potential avenue for future research to explore analyses encompassing manuscripts written in diverse languages. Furthermore, we have opened the door for potential comparative analysis, envisioning a juxtaposition of the WHO outputs with those of other prolific institutions, along with an examination of content related to the WHO.
What are the home messages?
We express our appreciation to the reviewer for raising this pertinent question. In response, we have undertaken a revision of our conclusion section, refining its articulation to underscore the pronounced influence of WHO initiatives on shaping the trajectory of research. This influence is evidenced not only by the growing volume of publications dedicated to NCDs but also by the alignment of these published topics with the prevailing and most significant NCDs, indicative of a synergistic and coordinated approach cultivated through collaborative efforts.
It could be better that lines 177-200 convert to the table.
We express our gratitude to the reviewer for providing this valuable suggestion. We have presented the distribution of editors across different continents as a comprehensive table, conveniently juxtaposed with a geographical map for enhanced visualization (see table 1).
The thickness lines legend of Figure 7-12 is unclear.
We thank the reviewer for raising this point. To enhance clarity, we have provided a comprehensive explanation at the outset of section 3.3. This clarifies that the thickness of the lines is directly proportional to the frequency of word repetition. In this context, a wider line signifies a higher frequency of the word's occurrence. Additionally, for each figure, we have included precise details about the frequency of the thinnest words. By providing this context, readers are equipped with the understanding that words represented by thinner lines exhibit lower repetition rates, while those with wider lines denote more frequent occurrences.
Does the author consider RTC study? Why did you not consider animal studies or others?
We thank the reviewer for the insightful suggestion. Within this study, our scope encompassed all publications related to NCDs, regardless of whether they were randomized controlled trials (RCTs) or not. Looking ahead, it would be valuable for subsequent research endeavors to specifically focus on RCTs conducted within the realm of NCDs, alongside exploring pre-clinical trials that address this particular category of diseases.
Round 2
Reviewer 1 Report
The authors have satisfactorily addressed my concerns.
Reviewer 2 Report
I would like to thank the authors for addressing my concerns and highlighting the need to bring into the dialogue research beyond that which is published in English. It is critical that expanding published research beyond that published in English is necessary to validate findings and expand our knowledge in this critical area of investigation.
Reviewer 3 Report
The author responds my comments.